# Seminal but not Serum Levels of Holotranscobalamin are Altered in Morbid Obesity and Correlate with Semen Quality: A Pilot Single Centre Study

**DOI:** 10.3390/nu11071540

**Published:** 2019-07-08

**Authors:** Jinous Samavat, Giulia Cantini, Maria Lorubbio, Selene Degl’Innocenti, Antonysunil Adaikalakoteswari, Enrico Facchiano, Marcello Lucchese, Mario Maggi, Ponnusamy Saravanan, Agostino Ognibene, Michaela Luconi

**Affiliations:** 1Division of Health Sciences, Warwick Medical School, University of Warwick, Coventry CV4 7AL, UK; 2Endocrinology Unit, Department of Experimental and Clinical Biomedical Sciences “Mario Serio”-University of Florence, 50139 Florence, Italy; 3Azienda Ospedaliero-Universitaria Careggi, 50134 Florence, Italy; 4Department of Biosciences, School of Science and Technology, Nottingham Trent University, Clifton, Nottingham NG11 8NS, UK; 5Santa Maria Nuova Hospital, 50122 Florence, Italy; 6Istituto Nazionale Biostrutture e Biosistemi (INBB), viale delle Medaglie d’Oro 305, 00136 Rome, Italy; 7Diabetes and Endocrinology Centre, George Eliot Hospital NHS Trust, College Street, Nuneaton, Warwickshire CV10 7DJ, UK

**Keywords:** active vitamin B12, cobalamin, morbid obesity, body mass index, sperm, seminal plasma

## Abstract

Vitamin B12 (cobalamin) is an essential cofactor in the one-carbon metabolism. One-carbon metabolism is a set of complex biochemical reactions, through which methyl groups are utilised or generated, and thus plays a vital role to many cellular functions in humans. Low levels of cobalamin have been associated to metabolic/reproductive pathologies. However, cobalamin status has never been investigated in morbid obesity in relation with the reduced semen quality. We analysed the cross-sectional data of 47-morbidly-obese and 21 lean men at Careggi University Hospital and evaluated total cobalamin (CBL) and holotranscobalamin (the active form of B12; holoTC) levels in serum and semen. Both seminal and serum concentrations of holoTC and CBL were lower in morbidly obese compared to lean men, although the difference did not reach any statistical significance for serum holoTC. Seminal CBL and holoTC were significantly higher than serum levels in both groups. Significant positive correlations were observed between seminal holoTC and total sperm motility (*r* = 0.394, *p* = 0.012), sperm concentration (*r* = 0.401, *p* = 0.009), total sperm number (*r* = 0.343, *p* = 0.028), and negative correlation with semen pH (*r* = −0.535, *p* = 0.0001). ROC analysis supported seminal holoTC as the best predictor of sperm number (AUC = 0.769 ± 0.08, *p* = 0.006). Our findings suggest that seminal rather than serum levels of holoTC may represent a good marker of semen quality in morbidly obese subjects.

## 1. Introduction

B12 (cobalamin, CBL) is a water-soluble vitamin synthesized only in bacteria [1]. Humans require an external source of this vitamin, primarily through dietary intake or from pharmacological supplementation, as it cannot be absorbed from the intestinal microbiota production [2]. This vitamin is crucial for homeostasis and growth as it acts as a coenzyme for the synthesis of methionine, a precursor of S-adenosyl methionine which provides the methyl groups for the methylation processes of macromolecules such as DNA, lipids, proteins, and neurotransmitters acting in the one-carbon-metabolism [3,4]. Around 10%–30% of circulating CBL is bound to the carrier transcobalamin in a complex named holotranscobalamin (holoTC), which is considered the bioactive vitamin B12 [5]. HoloTC has been proposed to represent a better marker than CBL of the vitamin deficiency status [6,7,8]. Another metabolic marker inversely associated with cobalamin is methylmalonic acid used to detect CBL deficiency at tissue levels [6], however, it is not routinely available for clinical testing.

Data present in the literature supports an association between CBL serum concentrations and metabolic pathologies, including obesity. Morbidly obese subjects [9,10,11] and patients with insulin resistance [12] present low concentrations of serum CBL. A positive association between CBL and dyslipidemia was found in two ethnic groups with type 2 diabetes (T2D) [13]. Similarly, an association was described between CBL and lower cord blood HDL in White Europeans [14]. A recent systemic review failed to establish any inverse association between circulating CBL and body mass index (BMI) when considering studies including not only obese but also underweight subjects [12]: this relation resulted in a J-shaped association. However, the very large heterogeneity in the meta-analyses considered in that study might have significantly affected the results [12]. Moreover, folate and age but not circulating CBL remained independently associated with the number of metabolic syndrome components in a large cohort of morbidly obese subjects following multivariate analysis [12]. Finally, the very recent analysis of the data from the nationwide population-based study in the U.S (National Health and Nutrition Examination Survey, NHANES), demonstrated the association of serum cobalamin with obesity by showing that those individuals with higher serum cobalamin levels were less likely to be obese [15].

It is known that morbid obesity is associated with sub-fertility [16,17,18,19], reduced semen quality [19,20,21,22], impaired sperm acrosome reaction [23], and sperm fatty acid composition and functions [24,25]. In particular, a large population study performed in men (*n* = 4,860), classified according to their BMI (underweight: *n* = 45, normal weight: *n* = 1,330, overweight: *n* = 2,493, obese: *n* = 926, and morbidly obese: *n* = 57), has shown that at both extremes of BMI (underweight and morbidly obese subjects), semen parameters were significantly worse and associated with lower sperm epididymal maturation compared to the other BMI classes [22]. Similar results have been found in a recent observational study conducted on 3,966 sperm donors in China for the underweight and overweight classes, but not for the obese class [26]. This may be due to enrolment bias (sperm donor bank) and the involvement of a small percentage of obese men (0.9%) (see also [19]).

Conflicting data is still present in the literature about the relation between low circulating CBL and male infertility [27,28,29]. However, a strong correlation was found between seminal CBL and sperm concentrations in a cohort of couples undergoing assisted reproductive techniques for infertility [30].

Altered absorption of CBL has been described in morbid obesity [9], however CBL and holoTC in serum and semen of these subjects have never been investigated and related to semen quality.

Therefore, the aim of the present cross-sectional observational study was to investigate the possible relationship between holoTC/CBL status and semen quality in a cohort of 47 morbidly obese versus 21 lean men. In particular, we compared holoTC versus CBL concentrations in serum and semen, and analysed their relationship with seminal parameters with no aim to use them for the diagnosis of vitamin deficiency.

## 2. Materials and Methods

### 2.1. Patients

Morbidly obese men who were on the waiting list for bariatric surgery at the Bariatric Unit of Careggi University Hospital in Florence (AOUC) were recruited to the proposed study. Any bias with infertility was avoided as the morbidly obese patients were not recruited in andrology centres. Inclusion criteria: BMI ≥ 38 Kg/m^2^, age between 20 and 65 years. Exclusion criteria: restrictive diet regimen, gastric balloon insertion, previous bariatric intervention, vitamin B12 supplementation, and tumour pathologies. A standard non-restrictive balanced diet had been prescribed to all patients by the Unit’s dieticians. Lean men were enrolled among subjects undergoing routine semen analysis for couple infertility at the Andrology Unit at AOUC. Inclusion criteria: 18.5 Kg/m^2^ ≤ BMI < 25 Kg/m^2^, age between 20 and 65 years. Exclusion criteria: couple infertility due to male factor, restrictive diet regimen, vitamin B12 supplementation, and tumour pathologies. The study was approved by the Local Ethical Committee and Institutional Review Board (approval protocol number 83/13 of 10.25.2003). All patients provided signed informed consent after receiving written and oral information on the study.

### 2.2. Biochemical and Anthropometric Measurements


1)Anthropometric measures: Height, weight, and waist circumference were measured in each subject and blood was drawn in the morning before seminal analysis.2)Vitamin B12: CBL and holoTC concentrations were measured in frozen serum and semen samples using the respective competitive and chemiluminescent enzyme immunoassays (ADVIA Centaur B12 and ADVIA Centaur Active-B12 Assays, Siemens Healthcare, Milan, Italy) based on LOCI technology on the Dimension Vista System (Siemens). Before holoTC determination, semen samples were diluted (1:3) with Multi-Diluent 13 (Siemens), centrifuged at 2,700g for 10 minutes and the supernatant was analysed. CBL and holoTC concentrations were expressed in pmol/L.3)Biochemical and sex steroid hormones: Glycated haemoglobin (HbA1c) was measured on the whole-blood samples by high-performance liquid chromatography ion exchange chromatography on a VARIANT II instrument (Biorad Laboratories, Milan, Italy). HbA1c values were used for the diagnosis of T2D at the 6.5% threshold. Serum levels of total testosterone (TT), sex-hormone-binding-globulin (SHBG), estradiol (E2), and the gonadotropin follicle-stimulating hormone (FSH) and luteinizing hormone (LH) were measured by immunoassay (Immulite 2000, M-Medical System, Italy); assay analytical sensitivity: 0.5 nM (TT), 55 pM (E2), 0.02 nM (SHBG), 20.1 mIU/ml (FSH) and 0.05 mIU/ml (LH). Free testosterone (cFT) was calculated on SHBG and TT, as previously described [31].


### 2.3. Semen Analysis

Human semen was obtained by masturbation according to the World Health Organization procedure [32] together with a blood sample on the same day or within one week. Semen parameters were analysed by a routine procedure [21,32]. Briefly, sperm morphology and motility were assessed by optical microscopy. Sperm morphology was evaluated as the percentage of normal and abnormal forms with Diff-Quick staining, scoring at least 100 spermatozoa/slide. Sperm motility was reported as the percentage of progressive motile, non-progressive motile, and immotile spermatozoa on at least 200 sperm/slide, total sperm motility was defined as progressive and non-progressive motile [21].

### 2.4. Statistical Analysis

Data was expressed as mean ± SD and as median (interquartile range, IQR) for normally and not-normally distributed parameters, respectively. All statistical analyses were performed on SPSS 24.0 for Windows (Statistical Package for the Social Sciences, Chicago, USA). Kolmogoroff–Smirnov’s test was used to determine the parametric distribution of data. CBL and holoTC data and most of the seminal parameters had a not-normally distributed distribution, while BMI, weight, and waist circumference were normally distributed. The Mann–Whitney U test was used for comparing two groups of not-normally distributed data, while the two-tailed Student’s t-test was applied for comparison of normally distributed data. Correlations were assessed using Spearman’s method for not-normally distributed parameters. ROC analysis was performed to evaluate accuracy, as well as sensitivity and specificity of the assessment of semen and serum holoTC and CBL concentrations in predicting semen quality. A *p* < 0.05 value was used for statistical significance.

## 3. Results

Anthropometric characteristics of morbidly obese and lean patients are reported in Table 1, along with the circulating levels of sex steroid hormones (TT, fT, E2, FSH, and LH), SHBG, and HbA1c. The prevalence of hypogonadism (TT < 12 nM) [33] and T2D (HbA1c ≥ 6.5%) was 66% and 28% in the morbidly obese cohort, while 22% and 0% in the lean cohort, respectively. Smokers and alcohol consumers are reported as percentages (Table 1).

CBL and holoTC concentrations were measured in the serum and semen of morbidly obese and lean subjects as shown in Table 2. CBL immunoassay had an analytical sensitivity of 37 pmol/L [34]. According to our measurements, the intra-assay and inter-assay coefficients of variation (CV%) in serum were 3.38 and 4.37 respectively, while in semen the values were 1.73 and 2.15, respectively. The values of analytical accuracy were evaluated according to CLSI EP05-A3 [35].

The ADVIA Centaur AB12 assay for holoTC in serum had the Limit of Detection (LOD) of 1.08 pmol/L and the Limit of Quantification (LOQ) of 5.00 pmol/L, determined as described in the CLSI documentation EP17-A2 [36]. The linearity was maintained for values 5.00–146.00 pmol/L, evaluated according to the protocol CLSI EP6-A [37]. According to our measurements, the intra-assay and inter-assay repeatability CVs% obtained in serum were 2.38 and 4.36 respectively, and in semen, 3.11 and 4.36, respectively.

CBL deficiency, defined as <148 pmol/L serum concentrations [38], was present in 8% of the morbidly obese men and not in the lean cohort.

HoloTC strongly correlated with CBL in both semen and serum in the whole population (Figure 1A,B). Conversely, only holoTC significantly correlated between semen and serum (*r* = 0.538, *r^2^* = 0.290, *P* = 0.014), while CBL did not.

In both cohorts, CBL and holoTC concentrations were significantly higher in semen versus serum samples (CBL: *P* < 0.001, and holoTC: *P* < 0.005, Table 2, Figure 2). Morbidly obese subjects presented significantly lower concentrations of both CBL (*p* = 0.006) and holoTC (*p* = 0.025) in semen compared to lean subjects (Table 1, Figure 2), while the differences between the two cohorts reached a statistical significance in the serum for CBL only (*p* = 0.012, Table 1, Figure 2). Not-statistically significant correlations were found between holoTC/CBL and BMI, weight, waist circumference (WC), sex steroid hormones, and HbA1c. 

The quality of seminal parameters was significantly reduced in the morbidly obese cohort compared to the lean subjects, in particular when considering total sperm motility, vitality, concentration, and total number, as well as semen volume and pH (Table 3). In order to investigate the relationship between CBL and semen quality, seminal characteristics and sperm parameters were evaluated in the whole population. A statistically significant positive association was found between seminal holoTC, CBL, and total sperm motility, concentration and total number, as well as a negative correlation with semen pH (Table 4). Conversely, no association was found between any parameter and holoTC and CBL in both semen and serum.

When the whole cohort of morbidly obese and lean subjects were stratified in two classes of holoTC concentrations, according to the median value of holoTC (118 pmol/L) derived from the seminal distribution of the whole cohort, total sperm number and seminal volume were significantly higher and pH significantly lower in those subjects with high seminal holoTC (Figure 3A–C). Similar results were obtained for total sperm number and pH (Figure 3D,E) but not for semen volume (Figure 3F) when subjects were stratified according to the median seminal CBL levels (1,069 pmol/L).

Finally, in order to compare the sensitivity and specificity of seminal and serum holoTC and CBL in predicting semen quality, a ROC analysis was performed. A statistically significant accuracy in predicting sperm number and concentration was found for seminal holoTC and CBL (Figure 4A–D), but not for serum values. The cut-off values identifying the best combination of sensitivity and specificity of the methods are indicated. These cut-off values were similar to the median values of holoTC and CBL distribution in semen (see Figure 3).

Table 5 reports the mean values of the different parameters when all subjects were stratified by the best cut-off values of seminal holoTC and CBL, as identified in the ROC analysis. Parameters of statistically significant differences between the two groups are shown in Table 5.

## 4. Discussion

We observed some interesting and novel findings in this study. In line with the higher holoTC and CBL concentrations found in semen compared to serum in the whole population of subjects studied, it is likely that CBL may concentrate from the circulation to the seminal compartment. The higher concentrations found for holoTC and CBL in semen might support a role of CBL in controlling semen quality. Accordingly, a recent review showed that CBL and antioxidant supplementation can improve semen quality, in particular enhancing sperm count and motility [27].

Secondly, we found that seminal CBL and holoTC concentrations were significantly lower than serum levels in morbidly obese compared to lean subjects. This suggests that alterations associated with high grades of obesity may somehow affect the process of local concentration in the reproductive tract, at least in the ejaculate, thus contributing to the reduced semen quality observed in morbidly obese men [17,18,19,20,21,22]. Indeed, we found a significant positive correlation between seminal holoTC/CBL and some seminal parameters, which lost statistical significance when serum holoTC/CBL concentrations were considered. In particular, this was found for both holoTC and CBL and total sperm motility, total number, concentration, and seminal pH. Conversely, a not-statistically significant correlation with normal sperm morphology was evident. This finding indicates an active role of seminal B12 in controlling semen quality at the post-testicular level, such as in the epididymis, where spermatozoa are concentrated and acquire active motility [39], as well as in semen production, acting on the prostate and seminal vesicles. Interestingly, alterations at the post-testicular level in the control of sperm maturation have also been hypothesized to contribute to the reduced semen quality found in morbid obesity [17,18,19,20,21,22]. The extreme obesity may affect sperm quality at a post-testicular level, which was confirmed by the absence of any effect on the normal morphology parameter and conversely related to testicular maturation as previously shown [19].

The correlation between morbid obesity and hypogonadism has been extensively validated [17,18,40], and the high prevalence of hypogonadism in this type of obesity has been further confirmed. However, in our study no significant correlations were found between seminal holoTC or CBL and sex steroid hormones, and no influence of hypogonadism on CBL was evident, suggesting that hypogonadism does not contribute to regulate holoTC and CBL concentrations.

The role of seminal CBL in the post-testicular control of semen quality was further suggested by the significant difference observed in total sperm number and semen pH between subjects stratified according to low and high holoTC and CBL. In particular, seminal holoTC, but not CBL concentrations, significantly differentiated patients for semen volume, which was substantially lower in subjects with low seminal holoTC. Interestingly, semen volume was one of the parameters strictly associated with prostate and seminal vesicle functionality [41]. Finally, according to the higher level of accuracy and best sensitivity/specificity associated with seminal holoTC evaluation compared to CBL, seminal holoTC turned out to be the best predictor of semen quality (sperm number and concentration), suggesting that this parameter could be considered as a potential useful marker of patients’ fertility.

One of the limitations recognized in our study was the limited number of subjects enrolled. Accordingly, our results should be considered as preliminary. However, the number of morbidly obese subjects enrolled was in line with the other studies conducted on morbidly obese subjects recruited in bariatric surgery programs [17,18,21,42,43,44]. To exclude that the difference in the number of subjects in the two cohorts could somehow affect the difference observed in holoTC and CBL, we calculated Cohen’s *d* values (Cd) for holoTC and CBL in semen and serum of the morbidly obese and lean groups. The values obtained (moderate level: semen holoTC Cd = 0.70, serum holoTC Cd = 0.77; high level: semen CBL Cd = 1.11, serum CBL Cd = 2.82) indicate that the small size of the groups was unlikely to affect the results. Our sample selection has two key strengths compared to previous papers [16,19,26]: a) absence of any recruitment bias – the morbidly obese subjects were not recruited in andrology centres, thus this does not affect fertility conditions, and b) parallel measurements of both holoTC and CBL concentrations in semen and serum of the same subjects.

## 5. Conclusions

In conclusion, our findings suggest that seminal rather than serum holoTC might represent a good marker of semen quality in morbidly obese subjects, whereby, despite the normal serum levels, low CBL and holoTC were found. Further studies investigating the local levels of holoTC on large populations including mildly obese/overweight subjects and elucidating the mechanisms which could affect semen quality are warranted.

## Figures and Tables

**Figure 1 nutrients-11-01540-f001:**
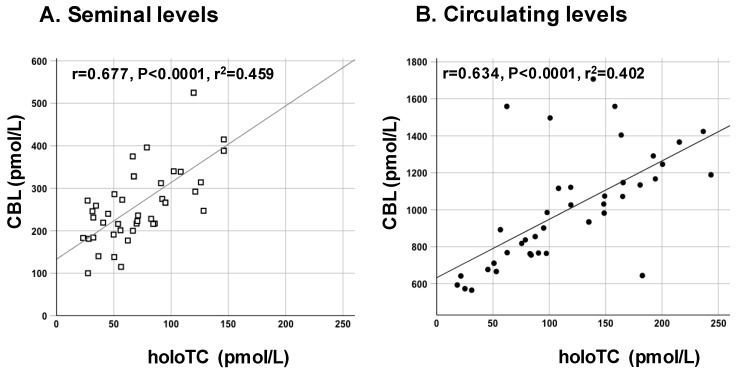
Positive correlations between CBL and holoTC concentrations in semen (**A**) and serum (**B**) in all subjects. Correlations were evaluated by Spearman’s test for not-normally distributed data in a univariate analysis: r, r^2^, and p from linear regression are indicated. CBL: cobalamin; holoTC: holotranscobalamin.

**Figure 2 nutrients-11-01540-f002:**
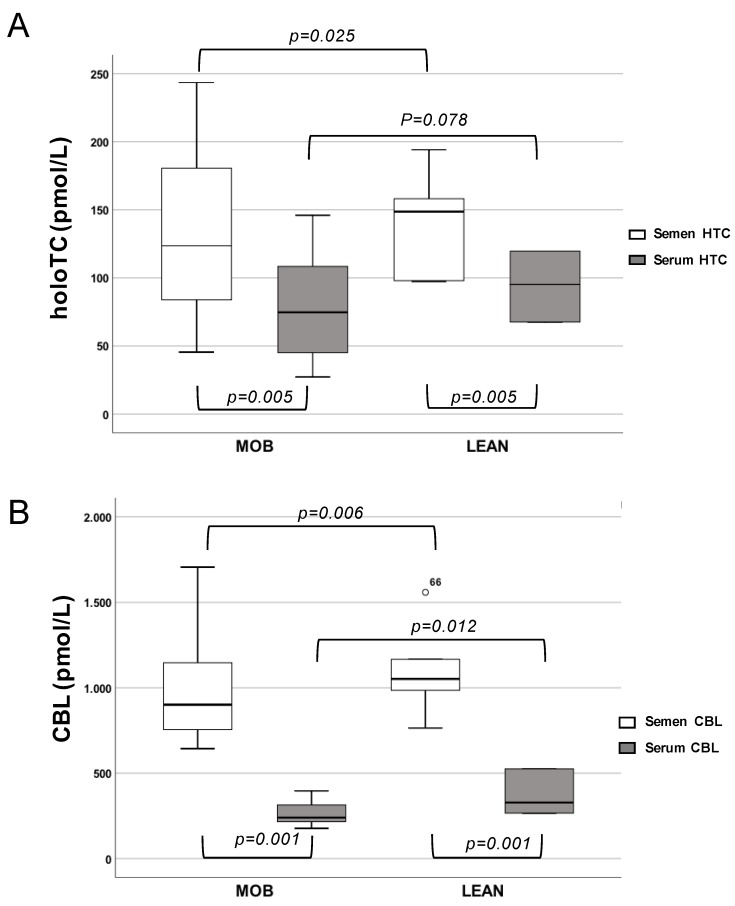
CBL and holoTC concentrations in serum and seminal samples from morbidly obese and lean subjects. Data is not-normally distributed and reported as box charts for holoTC (**A**) and CBL (**B**) serum and semen concentrations; p derived from the Mann-Whitney U test analysis is indicated. MOB: morbidly obese subjects; LEAN: lean subjects.

**Figure 3 nutrients-11-01540-f003:**
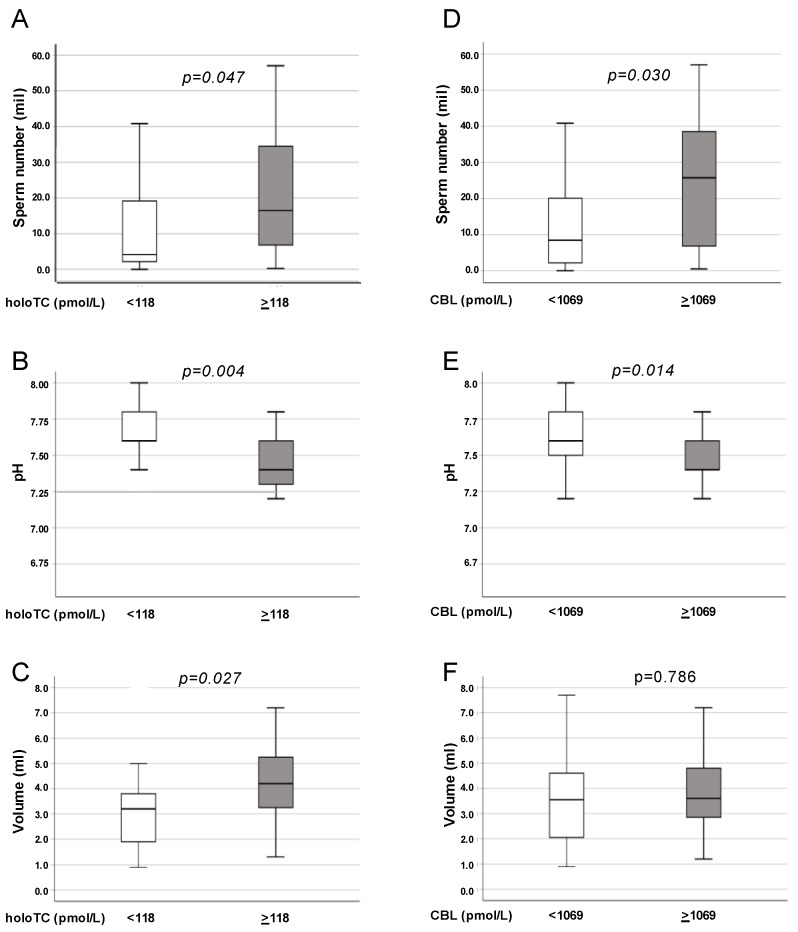
Seminal parameters stratified for seminal holoTC and CBL in the total population of obese and lean subjects. Data reported as box charts indicates sperm number (**A,B**), seminal pH (**C,D**), and volume (**E,F**) in all subjects stratified for low and high seminal holoTC (**A,C,E**) and CBL (**B,D,F**), respectively. Cut-off values were chosen as the median value of holoTC (118 pmol/L) and CBL (1,069 pmol/L) distribution in the whole population; p values were calculated with the Mann-Whitney U test.

**Figure 4 nutrients-11-01540-f004:**
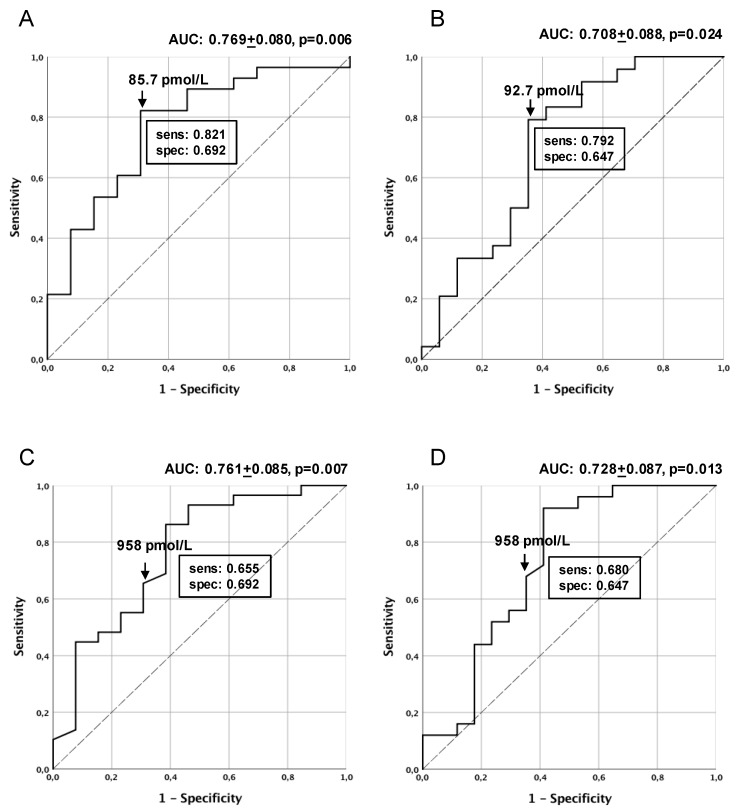
ROC analysis of seminal holoTC and CBL potency in predicting sperm number and concentration. ROC analysis of seminal holoTC predictive power for sperm number (**A**) and sperm concentration (**B**), and for seminal CBL predictive power for sperm number (**C**) and sperm concentration (**D**). Accuracy ± SD (AUC) and *p* value are indicated for each analysis, along with the cut-off value of holoTC and CBL associated with the best value of sensitivity (sens) and specificity (spec) of the method.

**Table 1 nutrients-11-01540-t001:** Biochemical and anthropometric characteristics of the patients: Anthropometric data and circulating levels of sex steroid hormones, SHBG and HbA1c are reported along with the prevalence of hypogonadism (TT < 12 nM), T2D (HbA1c ≥ 6.5), smokers (at least 1 cigarette/day), and alcohol consumers (at least 2 drinks/day) in each cohort. Data was normally distributed and reported as mean ± SD. Two-tailed Student’s t-test *p* values for comparison between obese and lean subjects are shown; χ^2^ was applied for comparison of non-continuous parameters (hypogonadism, T2D, smokers, alcohol consumers). Significant p values are indicated in italics.

	Morbidly Obese *n* = 47	Lean *n* = 21	*p*
Age (years)	42.0 ± 11.6	38.4 ± 7.9	0.207
BMI (Kg/m^2^)	46.5 ± 7.7	25.2 ± 3.9	0.0001
Weight (Kg)	149.5 ± 26.6	79.1 ± 10.0	0.0001
WC (cm)	142.7 ± 18.5	96.6 ± 4.0	0.0001
FSH (mIU/L)	4.47 ± 2.88	4.10 ± 1.47	0.740
LH (mIU/L)	2.98 ± 1.93	3.21 ± 1.86	0.757
TT (nM)	9.19 ± 4.31	14.44 ± 6.39	0.005
fT (nM)	0.219 ± 0.104	0.319 ± 0.104	0.015
E2 (pM)	139.2 ± 53.4	101.8 ± 35.2	0.024
SHBG (nM)	23.0 ± 9.8	28.3 ± 10.5	0.168
HbA1c (%)	6.2 ± 1.2	4.8 ± 0.3	0.0001
Hypogonadism (%)	66	22	0.029
T2D (%)	28	0	0.045
Smokers (%)	16	19	0.786
Alcohol consumers (%)	21	15	0.567

BMI: body mass index; WC: waist circumference; FSH: follicle stimulating hormone; LH: luteinizing hormone; TT: testosterone; fT: free-testosterone; T2D: type 2 diabetes.

**Table 2 nutrients-11-01540-t002:** CBL and holoTC concentrations in serum and semen: HoloTC and CBL concentrations are reported for morbidly obese and lean subjects as median (IQR) for not-normally distributed parameters and the Mann–Whitney U test was applied for comparison between obese and lean subjects. Statistical significance between seminal and circulating concentrations of holoTC and CBL: Significant *p* values are indicated in italics for comparison between morbidly obese and lean subjects.

	Morbidly Obese *n* = 47	Lean *n* = 21	*p* Mann–Whitney U
*Serum concentrations*			
holoTC (pmol/L)	64.5 (51.7)	95.1 (52.1)	0.780
CBL (pmol/L)	233.5 (99.2)	328.0 (259.0)	0.012
*Semen concentrations*			
holoTC (pmol/L)	83.9 (95.1) *	148.3 (67.0) *	0.025
CBL (pmol/L)	811.3 (477.0) **	1074 (456.0) **	0.006

CBL: cobalamin; holoTC: holotranscobalamin; * *p* < 0.005 and ** *p* < 0.001.

**Table 3 nutrients-11-01540-t003:** Seminal parameters: Seminal parameters of morbidly obese and lean subjects are reported as mean ± SD for normally distributed and median (IQR) for not-normally distributed parameters and analysed with the Mann–Whitney U test for comparison between obese and lean subjects. Significant *p* values are indicated in italics.

Seminal Parameter	Morbidly Obese *n* = 47	Lean *n* = 21	p Mann–Whitney U
Progressive Motility (%)	45.0 (31.0)	55.0 (19.5)	0.017
Total Motility (%)	52.0 (25.0)	65.0 (18.5)	0.004
Normal Morphology (%)	3.0 (5.0)	5.0 (6.0)	0.167
Vitality (%)	69.0 (29.0)	80.0 (6.0)	0.013
Total Sperm Number (millions)	34.5 (161.8)	220.0 (328.5)	0.001
Sperm Concentration(million/ml)	10.0 (52.6)	52.5 (75.0)	0.013
Volume (ml)	3.1 ± 1.7	4.2 ± 1.4	0.019
pH	7.6 ± 0.3	7.4 ± 0.2	0.001
Abstinence (days)	4.5 ± 3.8	3.8 ± 2.0	0.707

**Table 4 nutrients-11-01540-t004:** Correlations between seminal holoTC or CBL and seminal parameters. *r* and *p* values of Spearman’s correlation are indicated. The analysis was performed on the whole population.

	Seminal HoloTC	Seminal CBL
Total Sperm Motility (%)	*r* = 0.394 *p* = 0.012	*r* = 0.345 *p* = 0.027
Total Sperm Number (millions)	*r* = 0.401 *p* = 0.009	*r* = 0.482 *p* = 0.001
Sperm Concentration (mil/ml)	*r* = 0.343 *p* = 0.028	*r* = 0.465 *p* = 0.002
pH	*r* = −0.535 *p* = 0.0001	*r* = −0.481 *p* = 0.001

**Table 5 nutrients-11-01540-t005:** Stratification of the whole subject cohort according to seminal holoTC and CBL cut-off values identified by ROC analysis. Data is reported as median (IQR) for each parameter when all subjects were stratified in two groups according to the indicated cut-off values of seminal holoTC and CBL identified by ROC analysis. Significant *p* values after the Mann–Whitney U test are indicated.

	Seminal HoloTC			Seminal CBL		
	≥85.7 (pmol/L)	<85.7 (pmol/L)	*p*	≥958 (pmol/L)	<958 (pmol/L)	*p*
BMI (Kg/m^2^)	40.1 (39.0)	42.6 (15.8)	0.120	30.8 (27.1)	43.2 (36.1)	0.005
WC (cm)	120.0 (83.5)	142.0 (58.0)	0.032	120.0 (67.5)	144.0 (81.8)	0.018
Total Motility (%)	60.0 (79.0)	52.0 (75.0])	0.034	65 (61.0)	54.0 (75.0)	0.066
Total Sperm Number (millions)	166.5 (567.7)	30.8 (281.0)	0.001	162.4 (567.6)	41.6 (408.0)	0.018
Sperm Concentration (million/ml)	44.0 (306.0)	9.5 (70.0)	0.001	44.0 (178.1)	10.0 (213.0)	0.03
pH	7.4 (1.2)	7.8 (0.6)	0.0001	7.4 (1.2)	7.6 (0.8)	0.005

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
