# Peer review of "Seminal but not Serum Levels of Holotranscobalamin are Altered in Morbid Obesity and Correlate with Semen Quality: A Pilot Single Centre Study"

_nutrients, 2019, doi:10.3390/nu11071540_

Round 1
Reviewer 1 Report
The work presented by Samawal and team determined the total cobalamin and HoloTC in serum and semen collected from lean and morbidly obese participants. Also, the study presented data on the semen quality and how they might relate with cobalamin or holoTC.
The study design has not been clearly described; what was the type of clinical study? How were the study participants selected? Any inclusion and exclusion criteria? What were the limitations of the study design except for the low number of subjects? Other biomarkers related to B12 status were not presented, hence it was challenging to make a meaningful interpretation.
Detail comments are found below.
Intro
L 51 refer to Chemspider for an accurate terminology of cobalamin. Or keep the term simply as cobalamin.
L 53-55 the information presented here is not quite accurate and should be corrected.
L 56-58 the information is similar to what has been described in L 53-55, authors are advised to re-write and avoid the repetitive information.
L 63 MMA is a functional marker and complementary to other cobalamin markers. The claim of “most sensitive” is not accurate, even though a reference is provided.
L 89-94 The objective of the study has been somehow stated clearly. However, the preceding sentence about B12 status is quite disjointed.
Overall, the introduction is informative to set up the scene for the study.
Methodology
Section 2.1
What are the inclusion and exclusion criteria?
L 112-121 Data on the validation parameters should go to results.
Section 2.3
The protocol needs citations.
L 132 replace nonparametric distributed with the correct wording.
Results
Data on the patients’ characteristics should be presented at the beginning of Results section. What other biochemical markers were assessed besides anthropometric parameters? Did the study participants take metformin? What were the confounding factors?
Explain and justify the reason to use box plots to present the not-normally distributed data in Figure 2. Would the use of centiles be more appropriate for the data?
Consider this fact for the stats: the seminal data was more skewed than the serum data. Confounding factor MUST be included in the test. Also, the number of participants in the control group was less than the MO group.
L 177 replace the word “worse” with an accurate one fits for a scientific paper
Figure 3 Why did the authors use student t-test? Explain and justify this. Which are the control and MO groups represented by which plots?
Table 3
Explain and justify the use of Spearman correlation, because a statistical significance testing is not indicative of the strength of Spearman's correlation. Achieving a p<0.001 does not mean that the correlation is strong.
Discussion
L198-199 The result presented here is still premature and still needs further investigation and validation. Hence, the sentence “.........concentrated from the bloodstream to the seminal compartment in obese as well as lean subjects” should not be used instead it should be carefully expressed. The confounding variables were not considered, and other biomarkers related to cobalamin, such as folate (serum and rbc), homocysteine and MMA were not available to make any meaningful conclusion.
L 208-211 As per the previous comment about the Spearman’ correlations, authors should re-analyse the experimental data to prevent any interpretation that could potentially misleading.
L 212-214 This claim is conjecture, the current work only measure B12 in 2 different biological fluids.
L 228-229 Re-write the sentence for clarity.
Author Response
The attachment is the point-by-point response to the reviewer’s comments

Reviewer 2 Report
The authors reproted that both seminal and serum levels of B12ACT and B12TOT were lower in morbidly obese compared to lean men, although the difference did not reach any statistical significance for 42 serum B12ACT. Significant positive correlations between seminal B12ACT and total sperm motility, concentration and total number were found, as well as a negative correlation with semen pH (r=-0.535, p=0.0001). The authors findings suggest that seminal rather than serum levels of holotranscobalamin may represent a good marker of semen quality in morbidly obese subjects.
This paper is well written, and shed light on the one of the mechanisms of male infertility and possible therapeutic approach on male infertility.
Author Response

(The authors gave the same response as above.)

Reviewer 3 Report
This study investigated the relationship between vitamin B12 and semen quality in morbidly obese subjects. The results suggest that seminal rather than serum levels of active vitamin B12 may represent a good marker of semen quality in morbidly obese subjects.
Overall, the manuscript is well written. However, several issues should be addressed and improved.
1. The inclusion and exclusion criteria of obese and lean subjects who participated in the study should be described in detail. The backgrounds of 47 obese patients are unclear. The description in the limitation “morbidly obese subjects have not been recruited in andrology centers,” should be placed in the methods section. How about smoking status and representative complications such as diabetes or cardiovascular disease?
2. The authors claim that seminal levels of active vitamin B12 may represent a good marker of semen quality. However, there seems to be a difference in serum B12act between obesity and lean men (please provide the p value instead of “ns” in Figure 2 A), which may depend on the sample sizes investigated. As shown in Figure 1, B12act and B12tot were well correlated in both serum and semen. However, no information is provided about relationship of B12act and B12tot between semen and serum.
If a well correlationship is observed between semen variable and serum variables, I think that seminal as well as serum levels of active vitamin B12 may represent a good marker of semen quality.
If authors want to focus on the higher sensitivity of semen B12act than serum B12 or serum B12 tot for the semen quality, the authors should conduct ROC analysis and compare the results of AUCs.
3. Morbidly obese people may take unbalanced diet, which can lead to low intake of Vitamins including VitB12. If information for diet is unavailable, some explanations and discussion are needed.
4. In this study, data of other Vitamins are not described. Although the authors focused solely on VitB12, the levels of hormones including testosterone and other Vitamins such as B1, B2, and B6 can be also decreased in semen. Such information, if any, and discussion may be helpful particularly for the readers who are not familiar with such fields.
Author Response

(The authors gave the same response as above.)

Round 2
Reviewer 1 Report
The manuscript has been modified taking into account the comments of the reviewers. There are some minor corrections required before the manuscript can be fully accepted.
The introduction section could be improved in order to provide more accurate background information for the study, thus it needs revising.
Authors should be clear about the nutritional biomarkers of cobalamin, whether they are used for assessment of the status or for diagnosis. The function of holoTC proposed in the study and its correlation can be better communicated, so they are more reflective of the study. Even though Cohen test data was presented in the discussion, it has to be emphasised that results obtained from a study with over 40 morbidly obese participants should be carefully interpreted and generalisation of results is very discouraged.
Authors must be consistent in using the terms throughout the manuscript.
Seminal and serum levels should be replaced with concentrations where appropriate. Express as either serum or seminal holoTC is sufficient. Keep it simple.
Some detail comments are given here:
Intro
L 53 the sentence is inaccurate. Cobalamin is synthesised in the microbiota.
L 54-58 Write in simple sentences and correct the grammatical errors.
L 58-63 The expression and the content have to be corrected as they are not clear at all. Authors must decide which terms are to use between transcobalamin or holotranscobalamin.
L 60 HoloTC is the active cobalamin due to its bioavailability not because it’s absorbed by the cells. Please correct this.
L 62-63 truncated sentence, rewrite the sentence.
L 65 what was the significant association here?
L 66-69 a long sentence with double negatives should be avoided. Please write in simple language and concisely
L 93 ...the Word ‘forms’ is inaccurate here.
Methods
Exclusion criteria in morbidly obese subjects: were there any other chronic diseases that possibly had to be taken into account?
Table 2
Confirm the table content for accuracy. Check the significance levels.
L182 confirm the correlation is for the whole or the subset of the population?
L 256-259 rewrite the sentence by making it more simple and concise. The expression of both forms of vitamin B12 is not quite accurate. HoloTC is not a cobalamin form or vitamer.
Statistical
Describe the ROC test in the methods.
Table 4
Interpretation of the table content is needed in the body text. Mention this table also in text.
L 283 the study described in this manuscript is not a cohort study. Correct the text.
L 289 replace the word discriminate with differentiate
L 292-295 the sentence should be simplified. The result and potential implication should be better emphasised.
L 298-303 a very long sentence that would confuse readers. Write concisely and make the point effectively.
L 309 .....serum HTC would likely to represent ......
The result needs to be confirmed in future work, as such it’s not the absolute and careful interpretation is best here.
Author Response
We than the Reviewers for their additional comments. Below our reply.
REVIEWER 1
The manuscript has been modified taking into account the comments of the reviewers. There are some minor corrections required before the manuscript can be fully accepted.
1. The introduction section could be improved in order to provide more accurate background information for the study, thus it needs revising.
More information to better introduce the background on vitamin B12 and its relation with metabolic pathologies has now been added. The initial part of the section has been extensively rewritten (L53-75).
2. Authors should be clear about the nutritional biomarkers of cobalamin, whether they are used for assessment of the status or for diagnosis.
The use of cobalamin and holoTC as a marker of B12 status to be correlated with semen quality of these patients and not for the diagnosis of deficiency has now been clearly stated (L97).
3. The function of holoTC proposed in the study and its correlation can be better communicated, so they are more reflective of the study. Even though Cohen test data was presented in the discussion, it has to be emphasised that results obtained from a study with over 40 morbidly obese participants should be carefully interpreted and generalisation of results is very discouraged.
We are aware that the findings of our study are extremely preliminary, strictly related to morbid obesity and need further confirmation. Accordingly, we had introduced the term “pilot” associated to the “single centre study” in the title to highlight the preliminary aspect of the study. Moreover, we have further stressed this point in discussion (L300) and conclusions (L315-316).
4.Authors must be consistent in using the terms throughout the manuscript.
Seminal and serum levels should be replaced with concentrations where appropriate.
Done.
Express as either serum or seminal holoTC is sufficient. Keep it simple.
Done.
In addition, according to this Reviewer and the literature, we have changed the acronym of holotranscobalamin to holoTC throughout the MS.
Some detail comments are given here:
Intro
5.L 53 the sentence is inaccurate. Cobalamin is synthesised in the microbiota.
The sentence has been modified.
6.L 54-58 Write in simple sentences and correct the grammatical errors.
The sentences have been rewritten for clarity and errors corrected.
7.L 58-63 The expression and the content have to be corrected as they are not clear at all. Authors must decide which terms are to use between transcobalamin or holotranscobalamin.
According to the comment of this Reviewer in the previous revision, we decided to use cobalamin for total Vitamin B12 and holotranscobalamin for active Vitamin B12. We will use throughout the MS the acronym holoTC for indicating active vitamin B12, which is transported by transcobalamin.
8.L 60 HoloTC is the active cobalamin due to its bioavailability not because it’s absorbed by the cells. Please correct this.
The sentences have been re-written to “Around 10-30% of circulating CBL is bound to the carrier transcobalamin in a complex named holotranscobalamin (holoTC), which is considered the bioactive form of vitamin B12”.
9.L 62-63 truncated sentence, rewrite the sentence.
We apologize for the mistake that has now been corrected.
10.L 65 what was the significant association here?
It has been corrected into “positive association”
11.L 66-69 a long sentence with double negatives should be avoided. Please write in simple language and concisely
The sentence has extensively been rewritten.
12.L 93 ...the Word ‘forms’ is inaccurate here.
It has now been corrected.
Methods
13.Exclusion criteria in morbidly obese subjects: were there any other chronic diseases that possibly had to be taken into account?
Yes, they could have been, however we do not have enough data in our cohort for indicating them (for instance, hypertension), and when we decided the exclusion criteria we did not take them into consideration. Thus, we limited to the ones reported.
Table 2
14.Confirm the table content for accuracy. Check the significance levels.
Table 2 reports 2 types of significance: between morbidly obese and lean subjects (p MWU) and between serum and semen concentrations (indicated by * and **).
15.L182 confirm the correlation is for the whole or the subset of the population?
The correlation was obtained in the whole population (L188).
16.L 256-259 rewrite the sentence by making it more simple and concise. The expression of both forms of vitamin B12 is not quite accurate. HoloTC is not a cobalamin form or vitamer.
The sentences have been rewritten for clarity and the term “form” eliminated (L260-264).
Statistical
17.Describe the ROC test in the methods.
Done
Table 4
18.Interpretation of the table content is needed in the body text. Mention this table also in text.
We apologize for the wrong table numbers indicated in the text. Indeed, Table 4 was indicated with a wrong number in the Results (L221) and was already described in the previous version of the MS. Moreover, the meaning and impact of the correlations indicated in Table 4 had already been discussed in Discussion (L271-274).
19.L 283 the study described in this manuscript is not a cohort study. Correct the text.
We did not intend “ a cohort study” but a cohort of study. However, to avoid any confusion we changed “cohort of study” to the term “population” (L286).
20.L 289 replace the word discriminate with differentiate
Done
21.L 292-295 the sentence should be simplified. The result and potential implication should be better emphasised.
The sentence has been rewritten and simplified according to the Reviewer’s comment (L294-298).
22.L 298-303 a very long sentence that would confuse readers. Write concisely and make the point effectively.
The sentence has been rewritten and simplified (L302-307).
23.L 309 .....serum HTC would likely to represent ......
The result needs to be confirmed in future work, as such it’s not the absolute and careful interpretation is best here.
A sentence has been added to stress this point (L315-316). Moreover, the term “may” has been changed to “might” (L313).
Reviewer 3 Report
The manuscript has been greatly improved according to the comments.
Many abbreviations are observed in the manuscript. Are abbreviations such as MMA, WC, and HDL necessary?
Author Response
REVIEWER 3
Comments and Suggestions for Authors
1.The manuscript has been greatly improved according to the comments.
We thank the Reviewer for appreciating the improvement of the manuscript.
2.Many abbreviations are observed in the manuscript. Are abbreviations such as MMA, WC, and
HDL necessary?
When possible, they have now been avoided (for instance MMA, not for WC and HDL).